# Constructing the hierarchy of predictive auditory sequences in the marmoset brain

**Yuwei Jiang[1†], Misako Komatsu[2†], Yuyan Chen[1], Ruoying Xie[1], Kaiwei Zhang[1], Ying Xia[1], Peng Gui[1], Zhifeng Liang[1]\*, Liping Wang[1]\***

[1]Institute of Neuroscience, Key Laboratory of Primate Neurobiology, CAS Center for Excellence in Brain Science and Intelligence Technology, Chinese Academy of Sciences, Shanghai, China; [2]Laboratory for Molecular Analysis of Higher Brain Function, Center for Brain Science, RIKEN, Saitama, Japan

**Abstract** Our brains constantly generate predictions of sensory input that are compared with actual inputs, propagate the prediction-errors through a hierarchy of brain regions, and subsequently update the internal predictions of the world. However, the essential feature of predictive coding, the notion of hierarchical depth and its neural mechanisms, remains largely unexplored. Here, we investigated the hierarchical depth of predictive auditory processing by combining functional magnetic resonance imaging (fMRI) and high-density whole-brain electrocorticography (ECoG) in marmoset monkeys during an auditory local-global paradigm in which the temporal regularities of the stimuli were designed at two hierarchical levels. The prediction-errors and prediction updates were examined as neural responses to auditory mismatches and omissions. Using fMRI, we identified a hierarchical gradient along the auditory pathway: midbrain and sensory regions represented local, shorter-time-scale predictive processing followed by associative auditory regions, whereas anterior temporal and prefrontal areas represented global, longer-time-scale sequence processing. The complementary ECoG recordings confirmed the activations at cortical surface areas and further differentiated the signals of prediction-error and update, which were transmitted via putative bottom-up γ and top-down β oscillations, respectively. Furthermore, omission responses caused by absence of input, reflecting solely the two levels of prediction signals that are unique to the hierarchical predictive coding framework, demonstrated the hierarchical top-down process of predictions in the auditory, temporal, and prefrontal areas. Thus, our findings support the hierarchical predictive coding framework, and outline how neural networks and spatiotemporal dynamics are used to represent and arrange a hierarchical structure of auditory sequences in the marmoset brain.

**\*For correspondence:**
zliang@ion.ac.cn (ZL);
liping.wang@ion.ac.cn (LW)

[†]These authors contributed equally to this work

**Competing interest:** The authors declare that no competing interests exist.

## Editor's evaluation

This is an important primate study that combines fMRI, which can be used in humans, with systematic ECoG, which is not possible in humans. The work provides further support for a canonical neural model at the cortical level and provides new insight into a subcortical mechanism. The authors have responded well to the referees' methodological and expositional comments.

## Introduction

The mammalian brain is organized as a functional hierarchy, through which information propagates from the lower levels of sensory or motor regions (including subcortical areas) to the higher levels (e.g., temporal and prefrontal areas) (*Fuster, 1997*; *Huntenburg et al., 2018*; *Mesulam, 1998*; *Parras*

*et al., 2017*). At each processing level, neurons integrate information from multiple neurons at the level below, thus encoding increasingly abstract information over ever-larger temporal and spatial scales. Meanwhile, the reciprocal connections between cortical and subcortical areas provide neurons with feedback from the level above (*Felleman and Van Essen, 1991*; *Marques et al., 2018*; *Robinson et al., 2016*). In the time domain, the processing hierarchy is particularly important for cognition requiring a temporal integration of sensory stimuli and online action planning, such as spatial navigation and language (*Dehaene et al., 2015*; *Giraud and Arnal, 2018*).

The hierarchical predictive coding theory offers a unified framework for this functional hierarchy. It states that the brain develops a generative model of the world that constantly predicts sensory input (*Bizley and Cohen, 2013*; *Rao and Ballard, 1999*; *Spratling, 2010*). The comparison of predicted and actual sensory input then updates an internal representation of the world (*Keller and Mrsic-Flogel, 2018*). This process occurs throughout the cortical hierarchy. Specifically, the higher levels of the hierarchy send a top-down signal to lower levels in the form of a prediction of the bottom-up input to that area. The difference computed between the prediction and the bottom-up input is the prediction-error. The key assumption of the theory is that the prediction-error propagates across different depths of hierarchy, and in turn, the prediction propagates backward, providing signals to update the internal model at each level. Much of the research focuses on the signals of prediction and prediction-error at one unique level, such as the level of spatial detection in visual cortex (*Attinger et al., 2017*; *Fiser et al., 2016*) or one scale of temporal expectations in auditory cortex (*Gagnepain et al., 2012*; *Rubin et al., 2016*). However, the essential feature of predictive coding, that is, the notion of hierarchical depth (across multiple levels) is less well investigated (*Chao et al., 2018*; *Parras et al., 2017*).

Here, we combined whole-brain 9.4-T functional magnetic resonance imaging (fMRI) and large-scale electrocorticography (ECoG) recordings to gain both spatial and temporal neural information during a hierarchical local-global auditory sequence task (*Bekinschtein et al., 2009*) in the brains of marmosets (*Figure 1A and B*). These nonhuman primates are an important animal model for auditory processing because their social behavior and cognition are similar to those of humans (*Eliades and Miller, 2017*; *Miller et al., 2016*), and their ability of vocal learning from parents is comparable with vocal development in infant humans and song learning in birds (*Takahashi et al., 2017*). It is difficult to record neural activities (both low-frequency event-related potentials and high-frequency potentials, e.g., high gamma band) in the primary auditory cortex in humans and macaque monkeys because of the cortical sulci. As a result, studies on predictive processing in audition are restricted to electroencephalography (EEG) or ECoG recordings in associative cortical areas. By contrast, the smooth surface of the lissencephalic marmoset brain provides a unique opportunity to compare whole-brain fMRI and large-scale surface ECoG signals. In the present study, we used these combined techniques to study the distribution of activities over cortical and subcortical areas along the auditory pathway. We employed a local-global paradigm enabling us to measure predictive processing at one level along with the conditional propagation of prediction-errors to the next level, depending on the predictive context (*Figure 1B*). Both the blood-oxygen-level-dependent (BOLD) signals and the electrophysiological power changes were investigated across a broad frequency spectrum at the whole-brain level. The aims of the present study were to assess the hierarchical depth of predictive auditory sequences and to identify the neural networks and temporal dynamics for processing prediction-errors and prediction updates at two distinct levels: the lower local level with a shorter timescale (i.e., 150 ms, tone-onset asynchrony) and the higher global level with a longer timescale (i.e., 3 s, sequence-onset asynchrony).

## Results

### Local-global paradigm and a two-level hierarchical predictive processing model

We adopted the local-global oddball paradigm (*Figure 1B*), in which the auditory sequence comprised five identical tones (xxxxx) or four identical tones followed by a distinct tone ('xxxxY,' where x can be either tone of 800 Hz or 6000 Hz), which are referred to here as 'xx' and 'xY' sequences, respectively. During an initial habituation phase, five marmosets (three for the fMRI and two for the ECoG experiment) passively heard one sequence type (e.g., xY) in a given block. Then, during the following test phase, we probed for brain responses to novel sequences that either respected the habituated

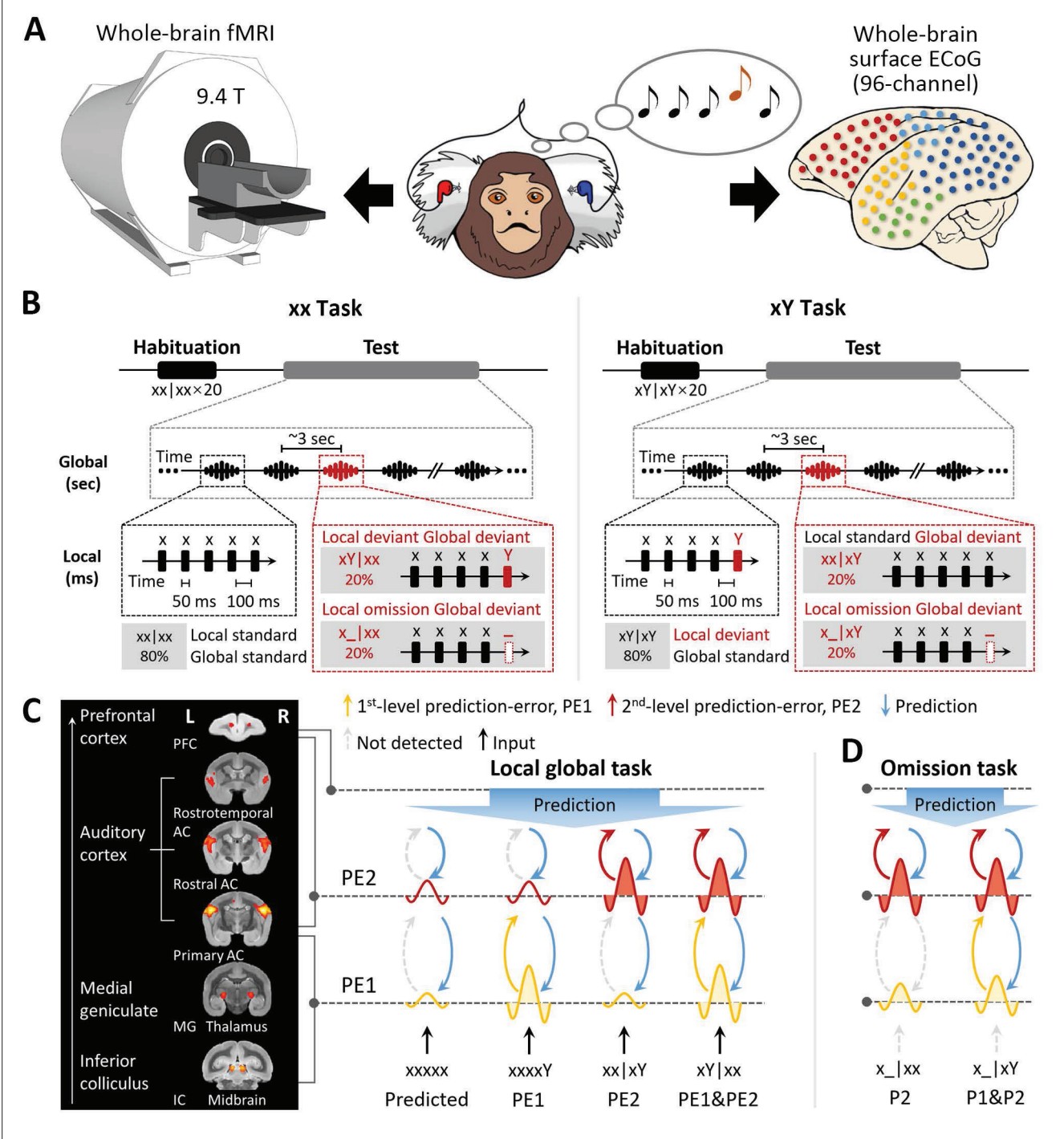

**Figure 1.** Experimental setup and local-global paradigm. (**A**) Experimental setup of whole-brain fMRI (9.4-T) and ECoG (96-channel); awake marmosets passively listen to auditory stimuli. (**B**) Schematic of local-global paradigm (for details, see Materials and methods). (**C**) *Left*, fMRI activations for auditory sounds relative to baseline, along the auditory pathway, in marmosets. *Right*, hypothesized predictive processing model of the local-global paradigm, corresponding to the auditory pathway. (**D**) Hypothesized predictive processing model of the local-global omission paradigm. P1, 1st-level prediction; P2, 2nd-level prediction; PE1, 1st-level prediction-error; PE2, 2nd-level prediction-error.

sequence (80% of trials) or violated it (20% of trials; alternative/deviant sequence [xx in this example]). In experiments assessing the effect of an omission, the 5th tone was simply omitted in the violation sequence (comprising 20% of trials) (*Figure 1B*; for details, see Materials and methods).

We used this paradigm to test the hypothesis of hierarchical neural dynamics during auditory sequence processing (*Figure 1C*), which predicts how the brain informs estimated statistics on the basis of the history of auditory sequences at two levels.

1. At the 1st (local) level, expectation violations result from tone-to-tone transition probability, which uses the most recent (150 ms) observations only, for example, the perception of an oddball (Y) tone deviating from four repetitions of standard (x) tones. The pattern of brain activation of predicted violation at this level is shown in *Figure 1C* (in yellow), where the Y tone induces significantly greater activation than the x tone in both xx and xY tasks, referred to as the 1st-level prediction-error (PE1).
2. These local sequences, whether normal (xx) or oddball (xY), are repetitively presented to form 2nd-order sequences (sequence to sequence), thereby generating expectations at a 2nd (global) level, which takes into account all five tones (~3 s). The 2×2 experimental design enables us to probe the neural responses to two types of global-level violations.
   a. Violation only at the 2nd level: an expectation violation results from the perception of the xx sequence deviating from the xY standard sequence (xx sequences in xY task, referred to as xx|xY). As shown in *Figure 1C* (in red), the xx violation sequences should induce significantly higher activation than the xY standard sequences, but they do not produce higher activation with the xx sequence at the local level. Here, the violation xx at the 2nd level is referred to as PE2.
   b. Violation at both 1st and 2nd levels: expectation violations resulting from the perception of the xY sequence deviating from the xx standard sequence (xY sequences in xx task, referred to as xY|xx) should generate two successive violations. Because the two levels of violations are activated sequentially, the 1st-level novelty (i.e., the 5th tone is Y) determines and triggers the 2nd-level novelty (the xY sequence), referred to as a PE1 and PE2 violation (*Figure 1C*).
3. The brain produces predictions at both local tone (Y) and global sequence (xx or xY) levels. When the incoming signal is omitted, brain responses should reflect solely the prediction signals and how they varied depending on the current context (xx or xY task) (*Figure 1D*). Specifically, in the xY task, where two successive oddball predictions are generated, we should observe a neural response to omission at both the 1st and 2nd levels, corresponding to the predictions of the Y tone and of the xY sequence. In the xx task, however, without the prediction of oddball tone in the xx sequence, the response to omission of the x tone may not be as strong as that of the Y tone at the 1st level, where only the prediction of the xx sequence at the 2nd level should be as robust as that in the xY task.

To test these hypotheses, we combined 9.4-T fMRI and 96-channel whole-brain ECoG of awake marmosets to assess the whole-brain neural networks and temporal dynamics during hierarchical auditory sequence processing, and searched for the neural representations of prediction-errors and predictions at two different levels.

## 1st-level (local) novelty (xY sequences)

To determine the activations of multiple violations, we performed a within-subject one-way analysis of variance (ANOVA), where the contrast images corresponding to different sequence types derived from a single session were used as within-subject factors (see Materials and methods). We first examined our data for the presence of a local mismatch response evoked by the deviance of the 5th tone. fMRI revealed the cortical regions activated by the novel tone, defined by the conjunction analysis using the contrasts between xY- and xx-evoked responses in both xx and xY tasks: (xY|xx – xx|xx) ∩ (xY|xY – xx|xY) (see Materials and methods). In the three marmosets, significant activation was found bilaterally along the auditory pathway, from the inferior colliculus (IC) in the midbrain and medial geniculate nucleus (MG) in the thalamus to auditory cortex (including auditory core and belt areas), the temporoparietal transitional area, dorsal prefrontal cortex, and anterior/posterior cingulate cortex (*Figure 2A*, red-yellow; $p<0.05$, false-discovery rate [FDR] corrected for multiple comparisons across the brain; *Supplementary file 1a*; individual marmosets: *Figure 2—figure supplement 1A*). These areas showed positive activation (relative to the mean of the baseline in this scanning run) for each auditory sequence and significantly higher activation for the Y tone in any xY sequence, regardless of the standard sequence pattern (xx or xY).

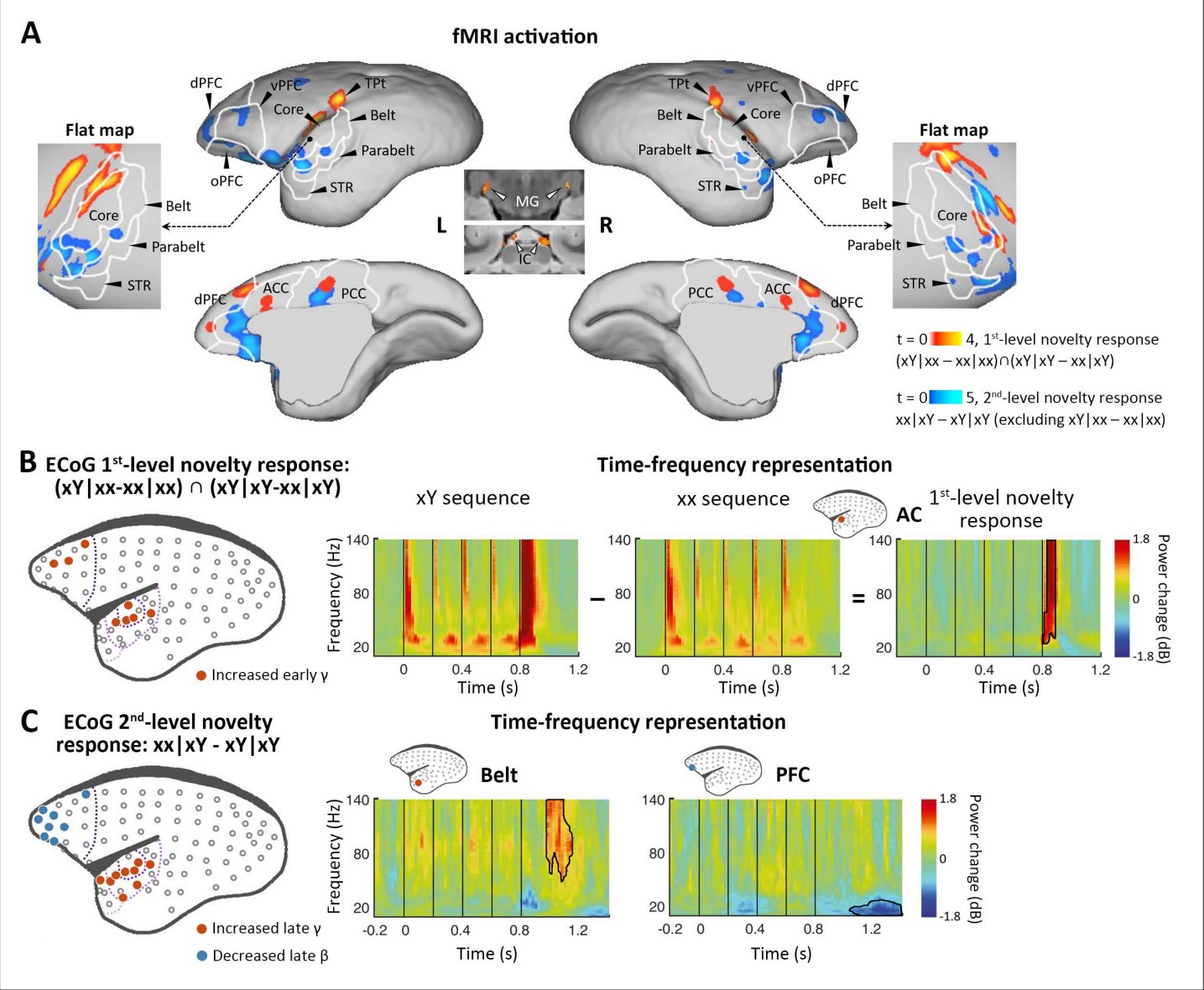

**Figure 2.** Neural activations to the 1st-level local and 2nd-level global novelties, respectively. (**A**) *Red-yellow*, group-level fMRI activations to 1st-level novelty identified by subtraction in both xx and xY tasks; *blue-light blue*, group-level fMRI activations to the 2nd-level novelty identified by subtraction only in the xY task (p<0.05, FDR-corrected). Color scales represent the t values. (**B**) *Left*, localization of ECoG electrodes for local novelties (marmoset J, p<0.05, cluster-corrected). The gray dots represent all 96 electrodes. The dotted lines from dark to light violet indicate frontal, auditory core, auditory belt/parabelt, and STR. *Right*, Time-frequency representation (TFR) of representative AC electrodes with 1st-level novelty response, generated by comparing local deviants (xY sequence) with local standards (xx sequence). The vertical lines indicate the onset times of five tones in each sequence. (**C**) *Left*, localization of ECoG electrodes for global novelties (marmoset J, p<0.05, cluster-corrected). *Right*, TFR of representative electrodes in the auditory belt cortex and PFC. ACC, anterior cingulate cortex; dPFC, dorsal prefrontal cortex; ECoG, electrocorticography; FDR, false-discovery rate; fMRI, functional magnetic resonance imaging; IC, inferior colliculus; L, left; MG, medial geniculate nucleus; oPFC, orbital PFC; PCC, posterior cingulate cortex; R, right; STR, superior temporal rostral area; TPt, temporoparietal transitional area; vPFC, ventral PFC.

The online version of this article includes the following figure supplement(s) for figure 2:

**Figure supplement 1.** Neural activations to the 1st-level local and 2nd-level global novelties in individual marmosets.

**Figure supplement 2.** Responses to different sequence types from electrodes with significant auditory response.

Because of the relatively poor temporal resolution of BOLD signals and the lack of time-frequency information with fMRI, as a complement, we performed ECoG (96-channel) recordings with the same auditory paradigm in two additional marmosets (one was referred to as marmoset J, and the other was referred to as marmoset N). The ECoG system acquires high-fidelity broadband neural signals from

an entire cortical hemisphere, with balanced spatial, spectral, and temporal resolutions. The spatial, spectral, and temporal dynamics of ECoG signals were quantified by the time-frequency representation (TFR) (see Materials and methods). A significant novelty response was detected by the corresponding comparison using a nonparametric cluster-based permutation test. The 1st-level novelty response was defined as a significant difference in TFRs for the 5th tone (between Y and x) in both xx and xY tasks. Consistent with the regional activation observed in the fMRI experiment, the electrodes (*Figure 2B* and *Figure 2—figure supplement 1B*, electrodes in red; p<0.05, cluster-corrected for multiple comparisons over frequency and time) exhibited an increase in γ-band power (>40 Hz) right after the 5th tone (peak at 74±34 ms after the 5th-tone onset) (*Figure 2B* and *Figure 2—figure supplement 1B*, right). In both marmosets, these electrodes responding to 1st-level novelty also displayed the highest responses to auditory sounds relative to the baseline (*Figure 2—figure supplement 2*; p<0.05, cluster-corrected), suggesting that the early γ-band power increase was most likely from the primary auditory cortex.

## 2nd-level (global) novelty (xx sequences in the xY task [xx|xY])

We next looked for a 2nd-level novelty response dependent on the overall sequence rather than individual tones. At this level, two internal sequence representations (xx and xY) were established during the habituation period, which enabled us to test the two different types of global novelties. We first tested the novelty responses in the xY task, where the deviant sequence (xx) only generated a violation at the 2nd level because it did not include local novelty. We then used fMRI to identify the brain regions showing significantly higher activation for xx sequences than for xY sequences in the xY task, and no difference between the xx and xY sequences in the xx task (i.e., when the sequence suddenly changed from xY to xx, but not from xx to xY). Unlike what was observed with the 1st-level novelty, we identified higher-order regions along the auditory pathway, especially in the temporal-prefrontal pathway and anterior-posterior cingulate areas, including bilateral anterolateral auditory area, superior temporal rostral area (STR), and prefrontal cortex (*Figure 2A*, blue-light blue; p<0.05, FDR-corrected; *Supplementary file 1b*; individual marmosets: *Figure 2—figure supplement 1C*).

ECoG showed that the 2nd-level novelty generated a later γ-band power (peak at 242±17 ms) after the 5th-tone onset than observed for the 1st-level novelty (*Figure 2C* and *Figure 2—figure supplement 1D*, electrodes in red; p<0.05, cluster-corrected). The extended γ-band power was from the electrodes located in the anterior auditory and superior temporal cortices. Furthermore, the frontal electrodes, in regions similar to those identified in the fMRI experiment, showed a late β-band power decrease (12–30 Hz) starting at 206±170 ms after the 5th-tone onset, with a longer latency and lasting for more than 300 ms (*Figure 2C* and *Figure 2—figure supplement 1D*, electrodes in blue; p<0.05, cluster-corrected).

Since the γ and β bands are thought to subserve bottom-up and top-down communications, respectively, in both humans (*Arnal and Giraud, 2012*; *Michalareas et al., 2016*) and monkeys (*Bastos et al., 2015*; *van Kerkoerle et al., 2014*), the ECoG results confirmed the activated regions shown by fMRI and, more importantly, functionally dissociated the fMRI activation patterns between frontal and temporal regions (*Figure 2A*). Combined with the finding of increased early γ-band activity shown in the 1st-level novelty response, we thus propose that the early and late increases in γ-band activity in the auditory and temporal cortices are associated with the bottom-up prediction-errors PE1 and PE2, respectively, and that the long-lasting decrease in β-band activity in frontal areas likely represents the subsequent top-down prediction or updating process.

## Violation at both 1st and 2nd levels (xY sequences in the xx task [xY|xx])

We next examined the violation sequence xY in the xx task, which generates novelty responses at two successive levels. In this analysis, the internal representation is for the xx sequence; thus, the conjunction analysis [(xY|xx> xx|xx) ∩ (xY|xY = xx|xY)] was used. That is, we searched for brain regions showing significantly higher activation for xY sequences than xx sequences in the xx task, and no difference between xx and xY sequences in the xY task. As predicted, the xY sequence produced a violation at both the 1st and 2nd levels, as the strongest activation during fMRI encompassed the lower-level auditory pathway, including the subcortical area MG, auditory core and belt regions, as well as the parabelt region of the higher-order auditory cortex and progressing to the temporal-frontal network,

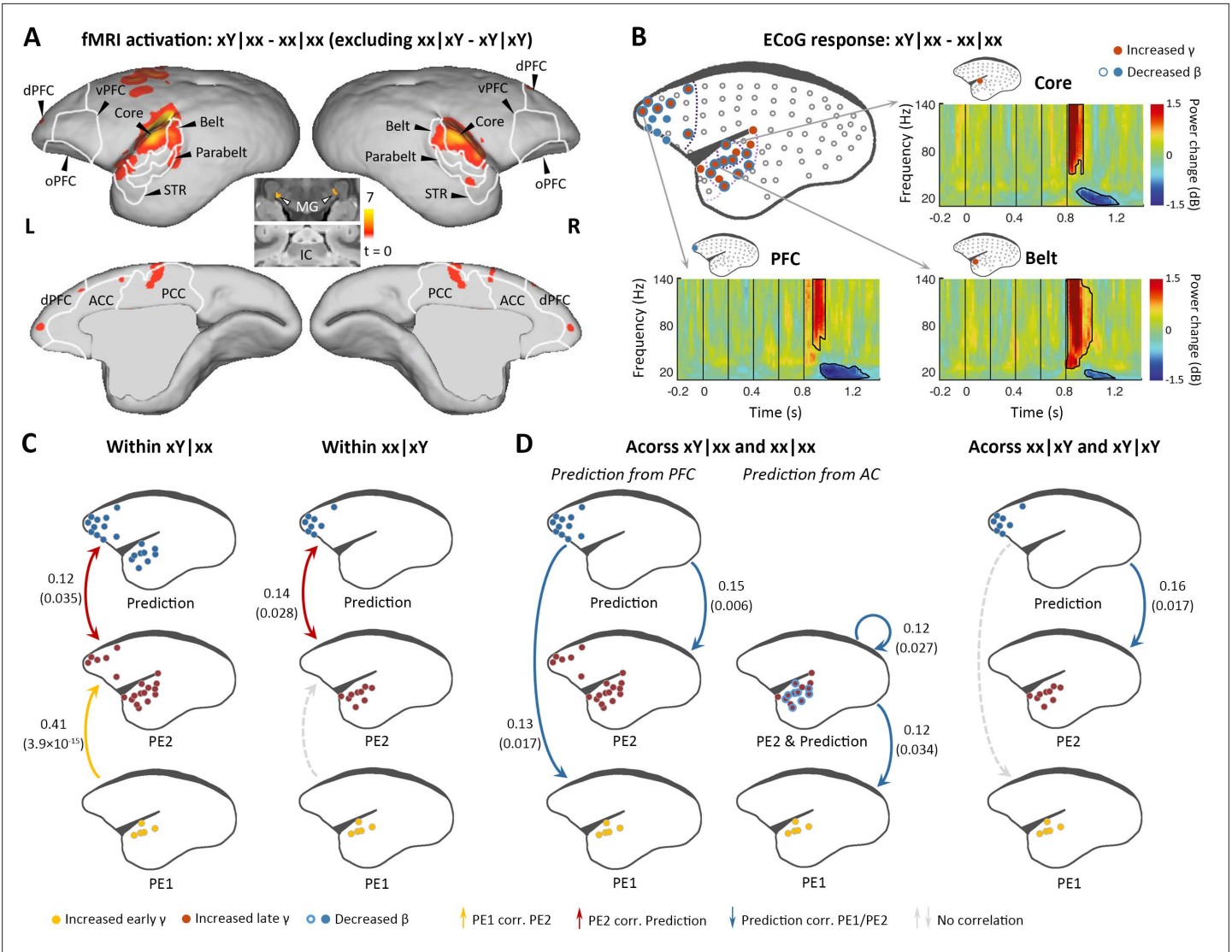

**Figure 3.** Brain activations to the two successive (local and global) novelties. (**A**) Group-level fMRI activations to both local and global novelties (p<0.01, FDR-corrected). Color scale represents the t values. (**B**) Significant ECoG responses detected by xY minus xx sequences in the xx task (marmoset J, p<0.05, cluster-corrected). (**C**) Diagram of the functional correlation among ECoG signals of 1st- and 2nd-level prediction-errors (PE1 and PE2, respectively) and prediction within deviants in the xx task (left, xY|xx) and xY task (right, xx|xY) (marmoset J). The color dots in the brain diagrams indicate the electrodes with significant responses found in corresponding comparisons, which were subsequently used in the functional correlation test (see Materials and methods). Lines represent significant functional correlations between signals from the paired brain diagrams. Labeled values close to lines provide the Pearson correlation coefficient (p value) of the corresponding correlations. Unidirectional arrows indicate relative temporal orders at which the signals appear, while bidirectional arrows indicate uncertain temporal orders of the signals. (**D**) *Left*, diagram of the functional correlation across trials, showing the correlation between prediction signals on xY|xx and PE1/PE2 signals on subsequent standards (xx|xx directly appeared after xY|xx). *Right*, the functional correlation across xx|xY and the subsequent xY|xY trials (marmoset J). (**D**) same format as (**C**). ECoG, electrocorticography; FDR, false-discovery rate; fMRI, functional magnetic resonance imaging.

The online version of this article includes the following figure supplement(s) for figure 3:

**Figure supplement 1.** Brain activations to the two successive (local and global) novelties in individual marmosets.

for example, STR, A8, and A10 (*Figure 3A*; p<0.01, FDR-corrected; *Supplementary file 1c*; individual marmosets: *Figure 3—figure supplement 1A*).

ECoG confirmed the activations along the auditory and temporal-frontal pathways, and most of the electrodes located at the surface of the superior temporal gyrus showed the early γ-band power increase, possibly denoting the representation of 1st-level prediction-error (*Figure 3B* and *Figure 3— figure supplement 1B*; p<0.05, cluster-corrected). The detection of the change from tone x to Y

signals a violation of sequence, and accordingly, the extended γ-band power increases (early and late γ-band power increases jointly lasting>200 ms) occurred in the anterior auditory and superior temporal regions (*Figure 3B* and *Figure 3—figure supplement 1B*, electrodes in red; p<0.05, cluster-corrected). In the marmoset J, the increased γ-band power also occurred for the prefrontal electrodes, suggesting a propagation of error signals from the 1st to the hierarchically higher 2nd level. Furthermore, both the frontal and auditory electrodes revealed decreases of the long-lasting β-band power (starting at 148±37 ms after the 5th-tone onset and lasting>300 ms) (*Figure 3B* and *Figure 3—figure supplement 1B*, electrodes in blue; p<0.05, cluster-corrected). Thus, these data suggest a potential two-level top-down modulation process of tones and sequences.

## Functional correlation between γ- and β-band power within and across trials

The proposed PE1/PE2 and prediction update represented by the early/late γ-band and long-lasting β-band power, respectively, were based on time-frequency characteristics. To verify a functional relationship between the frequency band and predictive function, we examined the directional correlation of corticocortical interaction in the novelty responses using the observed signals of PEs (γ-band activity) and prediction/updating (β-band activity) within and across trials (see Materials and methods). As shown in *Figure 1C*, the current framework predicts that the activation of the 1st-level violation should determine the activation at the 2nd level, especially on xY trials in the xx task. Thus, within a deviant trial, the hypothesis is that the early γ-band power increase (1st-level error) will trigger the late γ-band power increase (2nd-level error) and then interact with the late long-lasting β-band power decrease (prediction updating). Across trials, the β-band activity should update the internal representations of tones and sequences, which will affect the processing of subsequent trials. In this way, trial-by-trial fluctuations in the β-band power will affect the top-down predictions, and the amount of change will affect prediction-errors on subsequent trials.

We first examined the correlations between the three signals (early γ-band, late γ-band, and late β-band power) within the deviant trial xY|xx. Early γ-band (from primary auditory cortex, starting at 40 ms [marmoset J] and 60 ms [marmoset N] after the 5th-tone onset) and late γ-band (from auditory and superior temporal cortices, starting at 120 ms [marmoset J] and 160 ms [marmoset N] after the 5th-tone onset) prediction-error signals were positively correlated (*Figure 3C* and *Figure 3—figure supplement 1C*, left yellow), and late γ-band prediction-error and β-band (from auditory and prefrontal regions, starting at 140 ms [marmoset J] and 220 ms [marmoset N] after the 5th-tone onset) prediction-updating signals were positively correlated (*Figure 3C* and *Figure 3—figure supplement 1C*, left red). However, within the deviant trial xx|xY, we only observed a significant correlation between late γ-band and late β-band power (*Figure 3C* and *Figure 3—figure supplement 1C*, right) as hypothesized, as the 2nd-level error was not induced by the 1st-level error in the xY task.

Across trials, we investigated whether the long-lasting β-band signal in a deviant trial would affect the early/late γ-band signals in the subsequent trial, a standard (xx|xx) trial according to the experimental design. The prediction signal (β-band) from frontal cortex correlated significantly with early (PE1) and late (PE2) γ-band signals on the following xx|xx trial (*Figure 3D* and *Figure 3—figure supplement 1D*, left). Indeed, we also observed significant correlations between prediction signal from auditory cortex and PE1/PE2 signals from auditory cortex (*Figure 3D* and *Figure 3—figure supplement 1D*, left). By contrast, prediction signal from prefrontal cortex only significantly correlated with the late γ-band signal (PE2) from auditory cortex on the following xY|xY trial (*Figure 3D* and *Figure 3—figure supplement 1D*, right), suggesting a selective update of PE1 and PE2 based on different task contexts. These results demonstrate the sequential processing of the prediction-error signals from PE1 to PE2 (i.e., from primary auditory cortex to high-order auditory and superior temporal regions), which were propagated to frontal cortex where they interacted with prediction signals (*Figure 3C*); in turn, the prediction signals subsequently affected the PE1 and PE2 signals in response to new sensory inputs in the following trial (*Figure 3D*).

## Omission novelties (x_|xx and x_|xY) induce prediction signals dependent on task context

Finally, we examined omission responses. As proposed, omission of the 5th tone should reveal the brain's hierarchical predictions, and the observed omission response should vary according to the

expectation induced by the overall sequence context (xx or xY). fMRI was performed during sequences with omissions and during standard sequences in the xx or xY task (see Materials and methods). In the xx task, the omission induced the strongest activation in the posterior cingulate cortex, prefrontal cortex, and inferior temporal (TE3) regions (*Figure 4A*, red-yellow; p<0.001, uncorrected; *Supplementary file 1d*; individual marmosets: *Figure 4—figure supplement 1A*). However, responses to the omission in the xY task were found in the auditory belt cortex as well as the prefrontal and anterior cingulate cortices (*Figure 4A*, blue-light blue; p<0.05, FDR-corrected; *Supplementary file 1e*; individual marmosets: *Figure 4—figure supplement 1B*).

In the ECoG experiment, the omission effect was examined by comparing the omission sequence in the xx or xY task with the expected omission stimuli in the xxxx (four identical tones) block (see Materials and methods). Sequence x_|xY induced an early effect on γ-band power responses from 48±34 ms after the onset of the omitted tone for electrodes in the regions of primary auditory cortex that were similarly activated by the sequence xY|xx (*Figure 4B* and *Figure 4—figure supplement 1C*; p<0.05, cluster-corrected). The early latency of this peak response to the omission is consistent with the hypothesis that the response corresponds to an unfulfilled 1st-level prediction (*Figure 1D*).

In both xx and xY tasks, the omission also produced a significant effect on the γ-band power at a later time, lasting to 282±19 ms (x_|xx, starting at 151±38 ms) and 224±56 ms (x_|xY) after the onset of the omitted tone (*Figure 4C* and *Figure 4—figure supplement 1D*; p<0.05, cluster-corrected). The activated electrodes were located at associated auditory and superior temporal areas, which, as predicted, are similar to the activation areas where the 2nd-level (global) effect was observed in both xx and xY tasks (*Figures 2C and 3B*). Furthermore, in both tasks, the frontal and auditory electrodes revealed consistent long-lasting (>300 ms) decreases in β-band power, suggesting a feedback top-down updating of prediction in both contexts.

Finally, we compared the effect of omission between the xx and xY tasks, testing the unique role of prediction in the hierarchical predictive coding model. An omission was predicted to have a more profound effect on the xY task when a deviant stimulus was expected. In a region where two successive predictions are generated, we should observe a large response to omission, composed of superimposed activations corresponding to the prediction of the 'x' tone and the 'xY' sequence. By contrast, only one level of predictions should exist in the xx task; thus, the response to omission should be significantly smaller. The difference between omissions is shown in *Figure 5*. fMRI revealed greater activation in primary and associated auditory and prefrontal cortices during the xY task (*Figure 5A*; p<0.05, FDR-corrected; *Supplementary file 1f*; individual marmosets: *Figure 5—figure supplement 1A*). ECoG data similarly showed greater activation in the primary auditory and superior temporal cortices in response to the omission during the xY task than during the xx task (*Figure 5B* and *Figure 5—figure supplement 1B*; p<0.05, cluster-corrected, peaking at 85±19 ms after the 5th-tone onset). The early omission responses in the xY task reveal the PE1 signal generated by top-down prediction to actual auditory input.

Indeed, within deviant trials, we observed that the pattern of functional correlations for x_|xx and x_|xY were similar to xx|xY and xY|xx, respectively (*Figure 5C* and *Figure 5—figure supplement 1C*). That is, within the x_|xx trials, the PE2 only correlated with the prediction signal, and, within x_|xY trials, the PE2 correlated not only with the prediction signal but also with the PE1. Furthermore, functional correlation analyses across trials showed that in the xx task, the prediction in the x_|xx trial from frontal cortex correlated significantly with the PE2 in the subsequent trial (*Figure 5D* and *Figure 5—figure supplement 1D*, left), suggesting that the prediction signal is updated only at the 2nd-level sequence process. In the xY task, the prediction in the x_|xY trial from prefrontal cortex correlated significantly with both PE1 and PE2 in the subsequent trial (*Figure 5D* and *Figure 5—figure supplement 1D*, right), indicating that both 1st- and 2nd-level representations were updated by prediction signals from prefrontal cortex, a backward hierarchical top-down process. Moreover, the prediction in the x_|xY trial from auditory cortex only correlated with the PE1 in the subsequent trial (*Figure 5D* and *Figure 5—figure supplement 1D*, right), suggesting the 1st-level prediction signal from the auditory cortex.

## Discussion

We combined 9.4-T fMRI and high-density ECoG with a local-global auditory sequence paradigm to assess the hierarchical depth of auditory sequence processing in both spatial and temporal domains

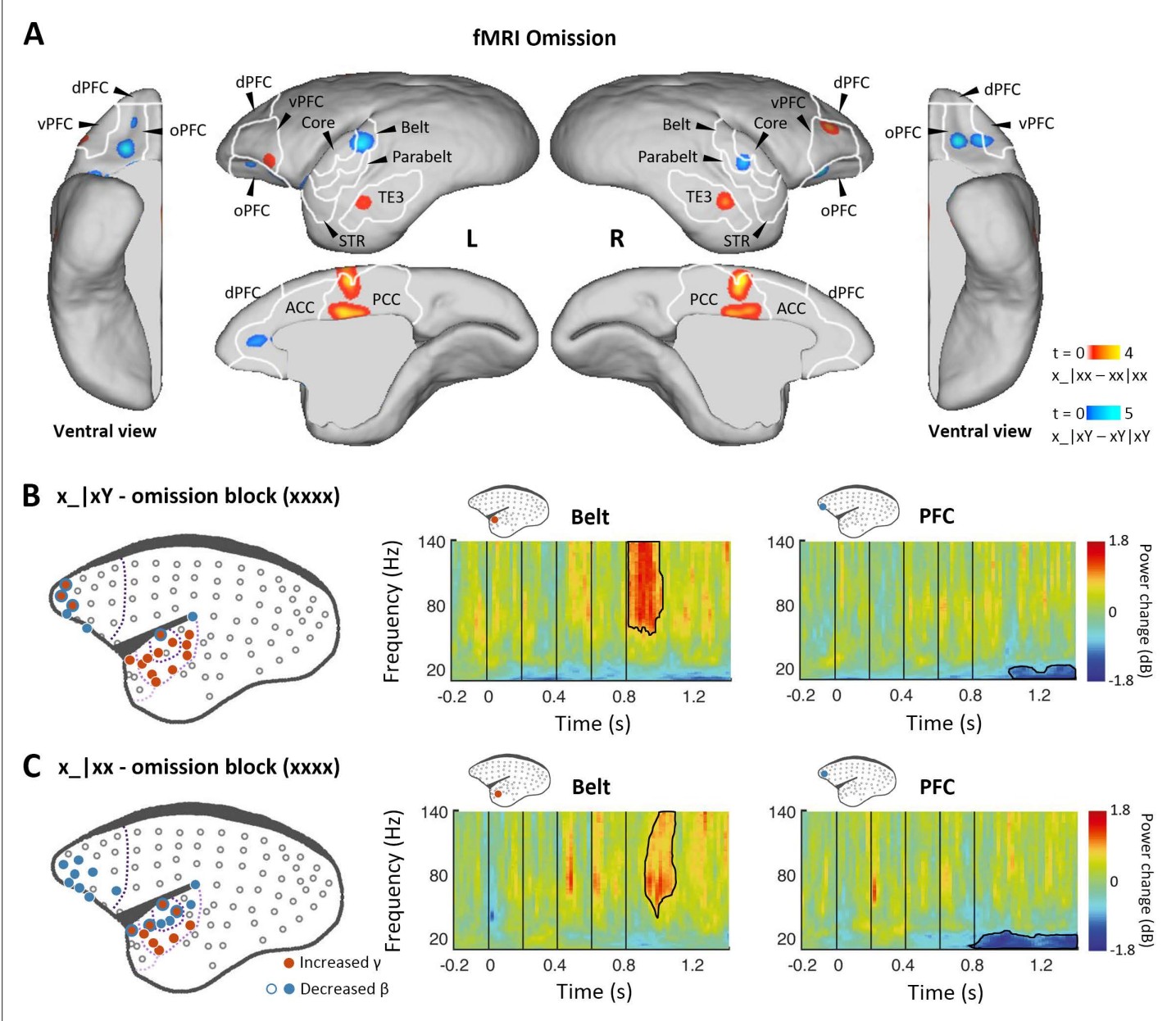

**Figure 4.** Prediction-error and prediction responses to omission of the 5th tone. (**A**) Group-level fMRI revealed areas responsive to an omission of the 5th tone in the xx (red-yellow, p<0.001, uncorrected) and xY tasks (blue-light blue, p<0.05, FDR-corrected). Color scales represent the t values. TE3, inferior temporal cortex. (**B**) *Left*, localization of ECoG electrodes with significant responses when comparing rare omissions in the xY task with frequent omissions in the omission block (consisting of xxxx sequences, where an omission is expected, marmoset J, p<0.05, cluster-corrected). *Right*, examples of respective electrode response in the auditory belt area and PFC. (**C**) Significant ECoG responses to the omission of the 5th tone when comparing rare omissions in the xx task with frequent omissions in the omission block (marmoset J, p<0.05, cluster-corrected), same format as for (**B**). ECoG, electrocorticography; FDR, false-discovery rate; fMRI, functional magnetic resonance imaging; PFC, prefrontal cortex.

The online version of this article includes the following figure supplement(s) for figure 4:

**Figure supplement 1.** Prediction-error and prediction responses to omission of the 5th tone in individual marmosets.

in the marmoset brain. At the whole-brain level, we found progressive encoding of information along the auditory pathway: local auditory information on a shorter temporal scale is encoded and propagated from the IC and MG, and primary auditory cortex to the posterior and anterior cingulate cortices, where the local 1st-level prediction-error (PE1) signal is characterized by the early γ-band. Global auditory information on a longer temporal scale propagates from primary auditory cortex

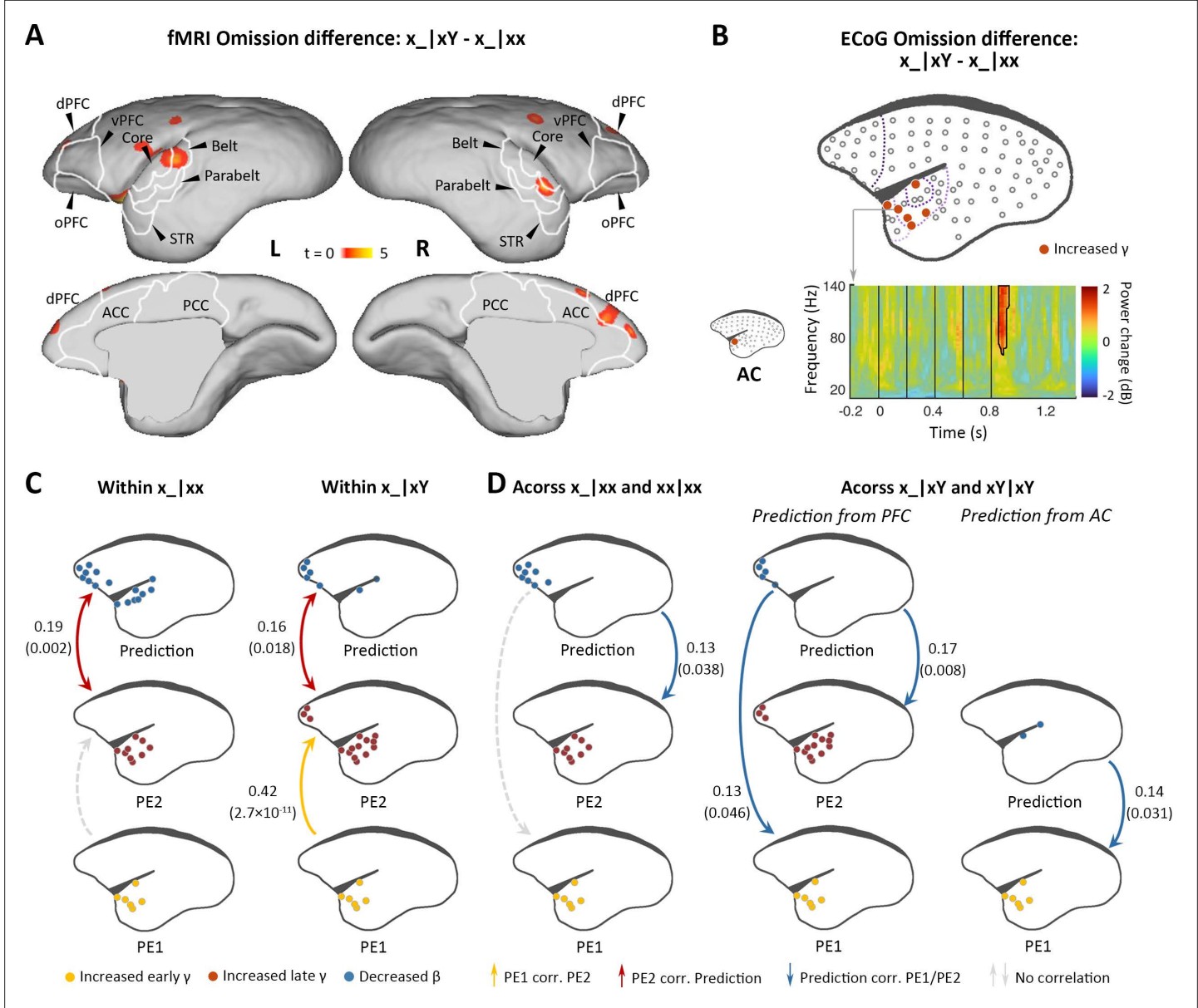

**Figure 5.** Difference in responses between local deviant omission and local standard omission. (**A**) Group-level fMRI revealed areas activated differently between local deviant omission (x_|xY) and local standard omission (x_|xx) (p<0.05, FDR-corrected). Color scale represents the t values. (**B**) *Top*, localization of ECoG electrodes with significant differences between x_|xY and x_|xx omissions. *Bottom*, an example of electrode showing significant early γ-band power increases in x_|xY omissions compared with that for x_|xx. (**C**) Diagram of the functional correlation results among ECoG signals within omission deviants in the xx task (left, x_|xx) and xY task (right, x_|xY) (marmoset J). The color dots in the brain diagrams indicate the electrodes with significant responses found in corresponding comparisons, which were subsequently used in the functional correlation test. Lines represent significant functional correlations between signals from the paired brain diagrams. Labeled values close to lines provide the Pearson correlation coefficient (p value) of the corresponding correlations. Unidirectional arrows indicate relative temporal orders at which the signals appear, while bidirectional arrows indicate uncertain temporal orders of the signals. (**D**) *Left*, diagram of the functional correlation results across trials, showing the correlation between prediction signals on x_|xx and PE1/PE2 signals on subsequent xx|xx trials. *Right*, the functional correlation across x_|xY and the subsequent xY|xY trials (marmoset J). (**D**) same format as (**C**). ECoG, electrocorticography; FDR, false-discovery rate; fMRI, functional magnetic resonance imaging.

The online version of this article includes the following figure supplement(s) for figure 5:

**Figure supplement 1.** Difference in responses between local deviant omission and local standard omission in individual marmosets.

to high-order auditory, superior temporal, and prefrontal cortices, where the 2nd-level prediction-error (PE2) signals are transmitted via the late γ-band. Furthermore, at both local and global levels, temporal and prefrontal cortices produce prediction feedback signals, which are transmitted in a long-lasting β-band. The prediction signals interact with the PE2 signals within a deviant trial as well as with the updated error signals in the subsequent trial. Furthermore, the neural responses to omissions in an auditory sequence at both local and global levels dissected the multiple top-down prediction systems from auditory and frontal cortices, respectively. The prediction signals dependent on the absence of bottom-up input selectively update the internal representations of tones or sequences, thus supporting the hierarchical predictive coding framework.

## Marmosets as an animal model for auditory sequences

To our knowledge, this is the first study to construct the hierarchy of predictive auditory sequences in the marmoset brain. Predictive coding was first articulated within the visual domain (*Rao and Ballard, 1999*). Investigating this with neural recordings in the auditory domain, especially in the primary auditory cortex, is more challenging that EEG or ECoG recordings are limited in associative cortical areas in humans and macaque monkeys (*Chao et al., 2018*; *Dürschmid et al., 2016*; *Wang et al., 2015*). However, the lissencephalic brain of the marmoset provides convenience to study primary and high-order auditory cortices (*Okano and Mitra, 2015*). While most of the previous marmoset studies were limited to electrophysiological techniques and mainly focused on auditory calls and pitch processing (*Bendor and Wang, 2005*; *Bendor and Wang, 2008*; *Toarmino et al., 2017*; *Zhou and Wang, 2012*), the present study has begun to examine the auditory sequences at the whole-brain level. In particular, we revealed that both prediction-error and prediction-updating signals are located in the core and belt regions of the auditory cortex. The results of whole-brain mapping of sequence processing would facilitate future studies in the marmoset brain, in which we can detail the finer representation of prediction and prediction-error signals in the auditory and related areas within the different cortical layers, even to the single-neuron level (e.g., via two-photon calcium imaging) (*Zeng et al., 2021*).

## Predictive signals in the subcortical areas along auditory pathway

Previous studies in humans and macaque monkeys have suggested that the local novelty could induce neural activations in the auditory cortex (*Chao et al., 2018*; *El Karoui et al., 2015*; *Uhrig et al., 2014*; *Wacongne et al., 2011*). However, due to poor spatial resolution of EEG and compromised signal-to-noise of fMRI (3T), the response to local novelties was barely described in the subcortical regions. Using 9.4-T fMRI in marmosets, we found the neural responses to the local novelty in the IC, MG, and primary auditory cortex. Researches using the classical oddball paradigm have shown considerable evidence for predictive processing in the auditory cortex (*Cornella et al., 2012*; *Dürschmid et al., 2016*) and lower subcortical nuclei, including IC and MG (*Grimm and Escera, 2012*). Furthermore, the prediction-error responses have been widely investigated in the subcortical auditory areas of humans (*Escera and Malmierca, 2014*) and animals (*Parras et al., 2017*), which support a large-scale hierarchical organization of auditory processing. Our finding is also consistent with a recent mouse study with electrophysiological recordings in these regions, where the authors examined local intervals (0.001–0.1 s) separating consecutive noise bursts and global rhythmic patterns (~1 s) of inter-burst interval sequences (*Asokan et al., 2021*). They showed that neural responses to local intervals were strong in the IC, but attenuated across MG and primary auditory cortex; the response to global rhythmic patterns was robust in primary auditory, but weak in MG and IC. Combined with the early peak latency of γ-band power for PE1, the local novelty responses in IC, MG, and primary auditory cortex are in line with the thalamocortical correlates of mismatch negativity (*Lakatos et al., 2020*), suggesting that classical oddball paradigms are involved in local prediction-error. Furthermore, our paradigm with the 2×2 design (e.g., xY|xx novelty) allows exploration of the interaction of local and global effects. Interestingly, in the xY|xx condition that the xx sequence as the internal template, the activations comprised both local and global effects were also observed in MG, suggesting that the auditory regions in the subcortical areas may also process the large timescale auditory information (*Bakay et al., 2018*; *Robinson et al., 2016*). Future studies using single or multiple units recording would help to clarify the function of sequence processing in MG and IC regions in marmosets.

## Hierarchical structure and two levels of predictive coding of auditory sequences

Studies using the local-global or human language paradigm have provided empirical evidence that temporal processing is hierarchical (*Chao et al., 2018*; *El Karoui et al., 2015*; *Uhrig et al., 2014*), with early sensory regions affected by the recent properties of the input stream and higher-order regions affected by the more complex features during longer time windows within a stimulus context (*Bekinschtein et al., 2009*; *Hasson et al., 2008*; *Lerner et al., 2011*; *Wacongne et al., 2011*). However, very few studies have examined the neural mechanism of the succession of prediction-error and prediction signals, that is, how lower-level or recent events trigger higher-level or prolonged signals. In one previous study, with data-driven analysis, *Chao et al., 2018* have clearly identified prediction and prediction-error signals at two different hierarchical levels and examined their interactions using high-density ECoG in macaque monkeys. The authors proposed full-global, partial-global, and no-global components in the two-level auditory sequence processing. However, similar to most other studies using the local-global paradigm (*Uhrig et al., 2014*; *Wacongne et al., 2011*), they tested the global violation by combining xx|xY and xY|xx novelties, which, in fact, contain two different types of predictions within the global violation. Therefore, our study for the first time separated the global novelties, that xx|xY novelty was only involved in the 2nd-level signal with the xY sequence as the internal sequence template, and the xY|xx novelty was involved in both 1st- and 2nd-level signals (the 1st-level novelty triggers the 2nd-level novelty), when the xx sequence was the internal template. As demonstrated in our hypotheses and results, the separation of the global novelty in the classic local-global paradigm was essential for understanding the neural mechanism of the depth of predictive auditory processing and the hierarchical propagation of prediction-error and prediction-updating information.

Finally, as most of the previous local-global studies did not test whether and how the prediction signals would respond to the omission in the auditory sequences, we adopted the omission sequence as a deviant to discern prediction signals at the two hierarchies. The neural activations induced by our deviant stimuli (e.g., xY|xx and xx|xY) cannot conclusively confirm the existence of prediction signals, because the novelty responses may act as modulatory signals of bottom-up input. However, the omission responses allowed us to study top-down prediction signal decoupled from bottom-up input signals, as without sensory input, the neural response to omission cannot be explained by any modulation of feedforward propagation and should contain specific information of upcoming predictive information (*Demarchi et al., 2019*). Thus, omission responses are considered as the proactive prediction and the violation of this prediction. Although detecting omission could happen retrospectively by comparing input and prediction template after the input is processed (*Bendixen et al., 2009*), our data demonstrated that responses to the omission only occurred after unexpected omissions (in contrast to the expected xxxx omission task), more likely reflecting the specific prediction signal rather than simple novelty detection (*Bendixen et al., 2009*; *Chennu et al., 2016*; *Fiser et al., 2016*). In the xx task, a fixed play rate of five identical tones ensures robust temporal expectations of the next tone. In line with previous works, the responses to rare omissions in the xx task may reflect a temporal prediction of the 5th tone and violation of the global sequence, regardless of the particular identity of the omitted tone. Our observation of x_|xx sequence only in temporal and frontal cortices using both fMRI and ECoG, may thus support that responses of prediction signal do not require bottom-up thalamocortical drive (*Demarchi et al., 2019*). In contrast, the responses of omissions could be found in xY task, only if both the timing and identity of the 5th tone could be predicted, because it was different from the previous four tones (*Sanmiguel et al., 2013*). The early γ-band increases and fMRI signals in auditory cortex, which were found by comparing x_|xY with x_|xx sequences, may reflect the unfilled specific expectation for identified stimulus because in both conditions temporal information was predicted. Therefore, since bottom-up inputs cannot account for omission responses, the early γ after following the omission onset in x_|xY sequence could be the consequence of 1st-level top-down prediction. The correlation we observed between the extended γ and long-lasting β signals in both x_|xx and x_|xY sequences implied the interaction between PE2 caused by global violation and 2nd-level prediction. The frontal prediction signal in x_|xY sequence for the update of both PE1 and PE2 on the following sequence and the temporal prediction signal for the update of the PE1, consistently supported the hierarchical top-down predictions.

In conclusion, our results from fMRI and ECoG recordings in marmosets may reveal a complete account of a hierarchical predictive coding framework of auditory sequences, including the prediction-error and prediction-updating information at two hierarchies.

## Materials and methods

**Key resources table**

| Reagent type (species) or resource | Designation | Source or reference | Identifiers | Additional information |
|---|---|---|---|---|
| Software, algorithm | Matlab | Mathworks (http://www.mathworks.com) | RRID:SCR_001622 | Version R2017b |
| Software, algorithm | Psychophysics Toolbox | Psychophysics Toolbox (http://psychtoolbox.org) | RRID:SCR_002881 | Version 3.0.12 |
| Software, algorithm | SPM Toolbox | Statistical Parametric Mapping (http://www.fil.ion.ucl.ac.uk/spm) | RRID:SCR_007037 | Version 12 |
| Software, algorithm | FieldTrip Toolbox | FieldTrip (http://www.fieldtriptoolbox.org) | RRID:SCR_004849 | |
| Software, algorithm | N-way Toolbox | N-way (http://www.modelslife.ku.dk/nwaytoolbox) | | Version 3.30 |

### Functional magnetic resonance imaging

Three adult common marmosets (*Callithrix jacchus*; two males, 350–420 g, 48–60 months) were used in the fMRI experiments. Awake marmoset fMRI training was adopted from *Silva et al., 2011*. Each marmoset was habituated to a simulated scanning environment over ~2 months. During training, the marmosets lay in a prone position with their heads immobilized by custom-made helmets. During routine fMRI scanning, the marmosets passively listened to auditory stimuli in a fully awake state without any active task. The auditory stimuli were presented using Matlab and Psychophysics Toolbox, and delivered through a pair of MR-compatible in-ear headphones (Model S14, Sensimetrics) at an average intensity of 75 dB SPL. The protocol of the fMRI study was approved by the Ethical Committee of the Institute of Neuroscience, Chinese Academy of Sciences (No. ION-20180522).

fMRI was performed with a 9.4-T Bruker BioSpec MRI scanner (Bruker, Billerica, MA), with an 8-channel phased array coil for marmoset brain imaging. The BOLD fMRI data were collected using a gradient-echo echo-planer imaging sequence with the sparse acquisition scheme so that a repetition time (TR) consisted of both acquisition time (TA, 1.5 s) and silence duration (2 s). The auditory stimuli were only played in the silence period to avoid any auditory interference. Under this scheme, the temporal resolution of the fMRI experiment was 3.5 s. The acquisition parameters for fMRI data were as follows: TR, 3.5 s; echo time, 15 ms; TA, 1.5 s, field of view, $40 \times 35$ mm$^2$; matrix, $80 \times 70$; voxel size, $0.5 \times 0.5 \times 1$ mm$^3$; number of slices, 33. A scanning run comprised 116 functional image volumes.

### ECoG implantation and recording

Two adult common marmosets (*C. jacchus*; marmoset J, male; marmoset N, female; 450–470 g, 42–90 months) were used in the ECoG study. Before the implantation of the ECoG array, the marmosets were familiarized with the experimental settings. The marmosets were then chronically implanted with whole-cortical 96-channel ECoG arrays in the epidural space. The arrays covered the left hemisphere of each marmoset, including most of the frontal, motor, parietal, temporal, and occipital cortices. The details of the surgical procedures for electrode implantation are provided in a previous study (*Komatsu et al., 2019*). The coordinates of recording electrodes were identified on the basis of the combination of pre-acquired MR images and postoperative computer tomography images.

ECoG signals were acquired using a Grapevine NIP system (Ripple Neuro, Salt Lake City, UT) at a sampling rate of 1 kHz. During the ECoG recordings, the marmosets sat in a sphinx position in an electrically shielded and sound-attenuated chamber with their head fixed. The auditory stimuli were delivered bilaterally by two audio speakers (Fostex, Japan) at a distance of ~20 cm from the head at an average intensity of 70 dB SPL. All procedures of the ECoG study were conducted in accordance with a protocol approved by the RIKEN Ethical Committee [No. W2020-2-008(2)].

### Auditory paradigm

A classical local-global paradigm was used in the fMRI and ECoG experiments (*Figure 1B*). For fMRI, each trial was a sequence of five tones. Each tone lasted 50 ms with an interval of 100 ms. Thus, the

total duration of the sequence was 650 ms and the interval of sequence onsets was 3.5 s. At the local level (millisecond timescale), a sequence of five identical tones comprised the local standard (e.g., xxxxx, referred to as 'xx'). When the 5th tone was replaced by a distinct tone, the sequence was identified as a local deviant (e.g., xxxxY, referred to as 'xY'). In the case of an omission, the sequence consisted of only four identical tones (5th tone omitted; e.g., xxxx). The frequency of the tone x or Y was either 800 Hz or 6000 Hz. At the global level (second timescale), a series of identical sequences played repeatedly were identified as global standards (e.g., xY). A distinct sequence presented rarely and randomly was identified as the global deviant (e.g., xx).

A given fMRI run started with a resting period of 14 s, following a habitation phase during which the selected global standard sequence was repeatedly presented 20 times to establish a global regularity. In the testing period, three blocks of 25 trials were presented sequentially, each followed by a 14 s rest. The 25 trials included 20 frequent global standards (80%) and five rare global deviants (20%). The global deviants were followed by more than one global standard. Each scanning run lasted 6 min 46 s. In the xx task, the testing blocks were either a mixture of 80% xx trials (xx|xx) and 20% xY trials (xY|xx), or a mixture of 80% xx trials and 20% omission trials (x_|xx). In the xY task, the testing blocks were either a mixture of 80% xY trials (xY|xY) and 20% xx trials (xx|xY), or a mixture of 80% xY trials and 20% omission trials (x_|xY). Each regular local-global session consisted of 3–4 runs of the xx task and 3–4 runs of the xY task, depending on the marmosets' performance on that day. In the omission local-global experiments, each session similarly consisted of 3–4 runs of the xx task and 3–4 runs of the xY task. The order of the tasks was random, and the frequencies selected for x and Y were balanced.

The duration of each ECoG trial was 850 ms, with each tone lasting 50 ms with an interval of 150 ms. The interval between sequence onsets was 3 s. The frequency used for tone x or Y was either 707 Hz or 4000 Hz. A separate recording run consisted of habituation and testing blocks with an omission sequence (xxxx) only. Each session contained 1–2 runs of the xx task with xY|xx or with x_|xx as the deviant, as well as 1–2 runs of the xY task with xx|xY or x_|xY as the deviant.

## fMRI data analysis

The fMRI data were analyzed with Statistical Parametric Mapping (SPM12, http://www.fil.ion.ucl.ac.uk/spm). Time series were slice-time corrected, and the acquired volumes were realigned to the first volume in the series. Data in which the head motion exceeded 0.5 mm or 0.5° were excluded. The realigned images were registered to the NIH marmoset anatomical template (*Liu et al., 2018*) and then spatially smoothed with a 0.6-mm full-width at half maximum Gaussian kernel.

For first-level analyses, a general linear model was established using experimental conditions (habituation, xx|xx, xY|xx, xY|xY, and xx|xY for regular local-global analysis or habituation, xx|xx, x_|xx, xY|xY, and x_|xY for omission local-global analysis), together with head motion parameters as regressors. At the single-session level, the statistical maps (threshold at p<0.01, uncorrected) generated by all auditory stimuli from a testing period without bilateral activation of auditory cortex were rejected. Therefore, the useful data for regular local-global stimuli comprised 55 sessions (21, 18, and 16 sessions for each of the three marmosets). The useful data for omission local-global stimuli were 48 sessions (18, 11, and 19 sessions for each of the three marmosets).

To determine the pathway of propagation of multiple novelty signals, a group analysis of within-subject one-way ANOVA was used for regular and omission local-global experiments, respectively. The single-session contrast images corresponding to each trial type derived from the 1st-level analyses were introduced to the 2nd-level ANOVA design in SPM. Subsequently, pair-wise t-tests were performed to compare the responses to different levels of novelties, which defined each type of contrast. To detect the auditory pathway of marmosets, activation in response to all sound stimuli (except during habituation trials) relative to that at rest was assessed using a threshold of p<0.001 (family-wise error rate corrected for multiple comparisons across the brain) with a cluster size of >50. To determine the novelty response of local level, we defined the contrast as a conjunction of xY|xx – xx|xx and xY|xY – xx|xY using a threshold of p<0.05 (FDR-corrected for multiple comparisons across the brain) and a cluster size of >5 (considering that the structural sizes of lower-level brain areas, like IC and MG, are relatively small, we chose the small cluster size for the local-level novelty activation). The contrast of global-level novelty response was determined using xx|xY – xY|xY (excluding xY|xx – xx|xx). The threshold used was p<0.05 (FDR-corrected) with a cluster size of >10. For both local- and global-level novelty effects in the xx task, the contrast was set as xY|xx – xx|xx (excluding

xx|xY – xY|xY) with a threshold of p<0.01 (FDR-corrected) and a cluster size of >10. To investigate the omission effect, the comparisons were x_|xx versus xx|xx using a threshold of p<0.001 (uncorrected) and a cluster size of >10, x_|xY versus xY|xY, and x_|xY versus x_|xx using a threshold of p<0.05 (FDR-corrected) and a cluster size of >10.

## ECoG data analysis

ECoG data were processed using Fieldtrip toolbox (*Oostenveld et al., 2011*). Trials were extracted from −1 to 2 s from the onset of the first tone. Each trial was resampled at 500 Hz and bandpass filtered using a bandpass filter order five with a window from 1 to 240 Hz. The electrodes with high-frequency noise over 60% of the trials were removed. For all remaining electrodes, trials with abnormal spectra and slow-wave activity were manually rejected (ft_rejectvisual.m). After artifact rejection, a recording run with fewer than five deviant trials was excluded. The useful data were notch filtered to remove the 50 Hz line noise, re-referenced using a common average reference montage, detrended to remove linear trends, and demeaned to apply baseline correction. The numbers of runs adopted were 23 (marmoset J) and 20 (marmoset N) for the xx task with xY|xx as the deviant, and 20 (marmoset J) and 18 (marmoset N) each for the xx task with omissions, xY task with xx|xY, and xY task with omissions.

Time-frequency analysis was conducted using 'mtmconvol' in Fieldtrip toolbox (ft_freqanalysis.m). Multitaper time-frequency transformation was based on multiplication in the frequency between 10 and 140 Hz (1 Hz step). The time from 300 ms before the 1st tone to 800 ms after the 5th tone (0.02 s step) was analyzed. Four time windows per frequency were analyzed, and the amount of spectral smoothing through multitapering was 0.5×the frequency. TFR was calculated by comparing deviant and standard trial types with baseline correction. To assess significant differences between TFRs, Monte-Carlo estimates of the significance probabilities were determined from the permutation distribution. The permutation was performed 1000 times by shuffling the trial labels 'deviant' and 'standard.' In each permutation, an independent-sample t-test was conducted at each frequency and time point. Then, a nonparametric cluster-based test was applied for the correction of multiple comparisons over frequency and time (ft_freqstatistics.m). The cluster-level statistics were computed by adopting the sum of the t values within each cluster and taking the maximum. The cluster-corrected threshold for significance was set at p<0.05.

To estimate the propagative function among different levels of prediction-error and prediction signals, the contributions of PE1, PE2, and prediction components were measured for each trial. The contribution of each component was achieved from parallel factor analysis using N-way toolbox (http://www.models.life.ku.dk/nwaytoolbox). According to the statistical results of each TFR comparison, the frequency and time domains of different components in each trial type were determined by the significant activity. The trial-by-trial TFR computed from the specific frequency and time domains was projected onto the spatio-spectro-temporal pattern. Then, the values for how much the PE1, PE2, and prediction components contributed to a given trial were obtained. To investigate the relationship among PE1, PE2, and prediction activities during deviant stimuli, the correlation between the contribution of one component derived from the corresponding significant electrodes and the contribution of another within the same type of trial were assessed. For example, the contribution of PE1 component was generated from the electrodes with significant response to local-level novelty. To observe the role of updating the prediction signal, the correlation between the contribution of the prediction component derived from deviant trials was further evaluated along with the contribution of the PE1 or PE2 component derived from post-deviant trials (which was the standard trial directly after a deviant trial). For the prediction component derived from the electrodes located at frontal cortex that with corresponding significant β-band activity, we calculated the correlation between PE1/PE2 component generated from all electrodes with corresponding significant γ-band activity and the prediction. However, for the prediction component elicited from the significant electrodes located at auditory cortex, we evaluated the correlation between PE1/PE2 component generated from auditory electrodes with corresponding significant γ-band activity and the prediction. If the prediction information was updated, the contributions of the prediction component to deviant trials would significantly correlate to the contributions of the prediction-error component to post-deviant trials. The correlation coefficient from a Pearson correlation with a p value of <0.05 was considered significant.

## Acknowledgements

The authors thank Yuri Shinomoto and Dr. Takaaki Kaneko for animal care and awake ECoG recordings. The authors also thank 9.4T high field small animal MRI platform and Primate physiology research platform from Institute of Neuroscience for data acquisition of awake fMRI. This work was supported by the National Science and Technology Innovation 2030 Major Program (2021ZD0204102 to LW), the Strategic Priority Research Program (XDB32070201 and XDB32030100 to LW, XDBS01030100 to ZL), the Pioneer Hundreds of Talents Program from the Chinese Academy of Sciences (to LW and ZL), the Shanghai Municipal Science and Technology Major Project (2018SHZDZX05 to LW and 2018SHZDZX05 to ZL), the National Natural Science Foundation of China (81801354 to ZL and 31900797 to YJ), the Youth Innovation Promotion Association Chinese Academy of Sciences (to YJ), the Brain/MINDS from the Japan Agency for Medical Research and Development (JP20dm0207069 to MK), and JSPS KAKENHI (JP19H04993 to MK).

## Additional information

### Funding

| Funder | Grant reference number | Author |
|---|---|---|
| National Science and Technology Innovation 2030 Major Program | 2021ZD0204102 | Liping Wang |
| Strategic Priority Research Program | XDB32070201 | Liping Wang |
| Strategic Priority Research Program | XDB32030100 | Liping Wang |
| Strategic Priority Research Program | XDBS01030100 | Zhifeng Liang |
| Pioneer Hundreds of Talents Program from the Chinese Academy of Sciences | | Zhifeng Liang<br>Liping Wang |
| Shanghai Municipal Science and Technology Major Project | 2018SHZDZX05 | Liping Wang<br>Zhifeng Liang |
| National Natural Science Foundation of China | 81801354 | Zhifeng Liang |
| National Natural Science Foundation of China | 31900797 | Yuwei Jiang |
| Youth Innovation Promotion Association Chinese Academy of Sciences | | Yuwei Jiang |
| Brain/MINDS from the Japan Agency for Medical Research and Development | JP20dm0207069 | Misako Komatsu |
| JSPS KAKENHI | JP19H04993 | Misako Komatsu |

The funders had no role in study design, data collection and interpretation, or the decision to submit the work for publication.

### Author contributions

Yuwei Jiang, Conceptualization, Data curation, Formal analysis, Funding acquisition, Investigation, Methodology, Validation, Visualization, Writing – original draft; Misako Komatsu, Data curation, Funding acquisition, Investigation, Resources, Validation, Writing – review and editing; Yuyan Chen, Kaiwei Zhang, Ying Xia, Investigation; Ruoying Xie, Formal analysis; Peng Gui, Conceptualization, Data curation, Formal analysis, Investigation; Zhifeng Liang, Data curation, Formal analysis, Funding

acquisition, Investigation, Project administration, Resources, Supervision, Validation, Writing – review and editing; Liping Wang, Conceptualization, Data curation, Funding acquisition, Methodology, Project administration, Resources, Supervision, Validation, Writing – original draft, Writing – review and editing

## Author ORCIDs
Yuwei Jiang http://orcid.org/0000-0002-9533-0760
Misako Komatsu http://orcid.org/0000-0003-4464-4484
Zhifeng Liang http://orcid.org/0000-0003-2758-1194
Liping Wang http://orcid.org/0000-0003-2038-0234

## Ethics
The protocol of the fMRI study was approved by the Ethical Committee of the Institute of Neuroscience, Chinese Academy of Sciences (no. ION-20180522). All procedures of the ECoG study were conducted in accordance with a protocol approved by the RIKEN Ethical Committee [no. W2020-2-008(2)].

## Decision letter and Author response
Decision letter https://doi.org/10.7554/eLife.74653.sa1
Author response https://doi.org/10.7554/eLife.74653.sa2

## Additional files

### Supplementary files
• Supplementary file 1. Details of fMRI results for brain regions of interest. (a): The peak fMRI activations of brain regions for 1st-level novelty. The image results were zoomed into human brain size for visualization and using MNI coordinates. IC, inferior colliculus; MG, medial geniculate nucleus; AuA1, primary auditory area; AuR, rostral auditory area; AuCM, caudomedial auditory area. L, left; R, right. (b): The peak fMRI activations of brain regions for 2nd-level novelty by comparing xx|xY with xY|xY. The image results were zoomed into human brain size for visualization and using MNI coordinates. AuAL, anterolateral auditory area; STR, superior temporal rostral area. (c): The peak fMRI activations of brain regions for both 1st- and 2nd-level novelty by comparing xY|xx with xx|xx. The image results were zoomed into human brain size for visualization and using MNI coordinates. AuRT, rostrotemporal auditory area; AuRTM, rostrotemporal medial auditory area; AuRPB, rostral parabelt auditory area; PE, parietal area. (d): The peak fMRI activations of brain regions for omission of the 5th tone in xx task by comparing x_|xx with xx|xx. The image results were zoomed into human brain size for visualization and using MNI coordinates. TE3, inferior temporal cortex. (e): The peak fMRI activations of brain regions for omission of the 5th tone in xY task by comparing x_|xY with xY|xY. The image results were zoomed into human brain size for visualization and using MNI coordinates. AuML, middle lateral auditory area. (f): The peak fMRI activations of brain regions for difference between local deviant and local standard omissions by comparing x_|xY with x_|xx. The image results were zoomed into human brain size for visualization and using MNI coordinates.

• Transparent reporting form

### Data availability
The fMRI and ECoG data that support the findings of this study are publicly available in Dryad: Jiang, Yuwei (2021), Constructing the hierarchy of predictive auditory sequences in the marmoset brain, Dryad, Dataset, https://doi.org/10.5061/dryad.j3tx95xfp.

The following dataset was generated:

| Author(s) | Year | Dataset title | Dataset URL | Database and Identifier |
|---|---|---|---|---|
| Yuwei J | 2021 | Data from: Constructing the hierarchy of predictive auditory sequences in the marmoset brain | http://dx.doi.org/10.5061/dryad.j3tx95xfp | Dryad Digital Repository, 10.5061/dryad.j3tx95xfp |

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
