## [Editor Report]

This is an important primate study that combines fMRI, which can be used in humans, with systematic ECoG, which is not possible in humans. The work provides further support for a canonical neural model at the cortical level and provides new insight into a subcortical mechanism. The authors have responded well to the referees' methodological and expositional comments.

---

## [Decision Letter]

**Decision letter after peer review:**

Thank you for submitting your article "Constructing the hierarchy of predictive auditory sequences in the marmoset brain" for consideration by *eLife*. Your article has been reviewed by 2 peer reviewers, and the evaluation has been overseen by a Reviewing Editor and Andrew King as the Senior Editor. The reviewers have opted to remain anonymous.

The work combines fMRI with extensive ECoG coverage to provide support for a hierarchical mismatch organisation. Such an organisation has been suggested by previous work, as have the β prediction and γ mismatch signals. The work has demonstrated a pattern of regional organisation in this species that is very suitable for further evaluation at the neural level.

The referees were both enthusiastic about the work, but have made detailed comments about the exposition in their evaluations below that we expect you to address in the final version of the manuscript.

*Reviewer #1 (Recommendations for the authors):*

Line 37: Putative instead of putatively.

Line 42: Neural circuits. I really read circuits to represent individual neuronal connections, or at the very least, small local neural pathways. As an fMRI and ecog study, that kind of resolution isn't available. I would recommend using neural networks.

Lines 49-58: There is a great deal of discussion here about cortical levels. However this study includes data from the thalamus and IC, both of which are generally considered to be subcortical. I would suggest the authors define their terms more clearly, either by just using the term auditory pathway, or higher.

Line 78: Remove increasingly. Don't sell your model short, it is an important animal model and you tell us why right afterwards.

Line 78: I'm not sure exactly what the final layout will look like, but try to keep the figure close to the first reference to it. It's 4 pages after the fact right now, and I had to go looking.

Lines 77 and 78. Local-global is mentioned several times by this point, but is not defined nor is there a direct reference until 100 when it is finally detailed. Please put a reference on the first mention of it.

Line 94: Again I prefer networks to circuits for this kind of macro level investigation.

Line 95. Use a shorter time scale, not a short time scale. 150ms isn't short compared to the response of a neuron. By keeping your terms relative, people like me can't complain.

Lines 101-105, and especially the word alternatively at 103. The description of the stimuli is unclear. I cannot tell if x can be either 800 or 6000Hz, so that there are stimuli of 800 800 800 800 800 and 800 800 800 800 6000 as well as 6000 6000 6000 6000 6000 and 6000 6000 6000 6000 800 or only the former. Some rephrasing should easily tidy this up.

Line 153: PE1 and PE2 need to be defined in the caption text, not just in the figure image.

Line 246: None of the MG or the cortical areas are "low-level auditory pathway". You could maybe get away with lower-level and keep it relative in the context of this study, but even then, you have IC which is lower than the MG, is not mentioned here, and is still not low-level as it's midbrain not brainstem. I'd recommend saying lower cortical and subcortical areas, or something to that effect. Be very aware that the IC does a lot of auditory processing, and probably is most closely analogous with V1 in the visual system, so this is the kind of error that can upset a lot of researchers.

Line 257: It took me quite a while to figure out what you meant by In the marmoset (J). Remember, we didn't do your study, so we don't know your labelling and won't because most people won't read your supplemental section. Say: In the marmoset individual, referred to as J, the increased…

Line 313: Remove neural. It implies you have a resolution that you don’t have.

Line 354: Monkey should be marmoset, be consistent in how you refer to your animals. Also, clearly indicate that J is the animal's identifier, and be consistent with that throughout the paper.

Line 363: Prediction not predictions.

Line 376: "suggesting that the prediction signal is updated on at the 2nd level sequence process". This is one of the most important results of the paper and is buried mid paragraph here. Show it off a bit more, be proud of your results, they're good.

Lines 396 and 399: short and long temporal scales. I would really recommend defining your terms, calling them the shorter and longer time scales, and specify their lengths when you define them. Again, ecog and fmri just don't have what I would call short temporal resolution.

Line 412: Induce not induced.

Line 417: Mouse not mice.

Line 420: Either "*The* neural response" or "neural *responses* to local intervals *were*".

Line 421: Declined is awkward here. Try attenuated.

Line 427: Allows exploration of the…

Line 429: Subcortical areas instead of regions, to keep from repeating regions twice in the space of 5 words.

Line 429: Bakay et al. 2018 and Robinson et al. 2016 both suggest this is likely as well, should probably reference them.

Line 447: Remove neural. It's superfluous.

Line 465: Tones not tone.

Line 469: May thus support that responses to prediction signals do not require.

Line 473:.…… auditory cortex, which *were* elicited by…

Also, elicited is a 5 dollar word and really awkward in this context, I'd just use found.

Line 476: The early γ after following *the* omission onset.

Line 482: Predictions. In conclusion our results from fMRI and ECoG recordings in marmosets reveal an account of hierarchical…

Also, I'd recommend separating that in conclusion sentence out into its own paragraph.

*Reviewer #2 (Recommendations for the authors):*

1) Why are MGB and IC responses not shown in Figure S1B, and C? No activation would presumably also be interesting. Currently, it is unclear whether they were not activated, or whether it was not measured.

2) Figure 2A. Please explain t = 0…4 (we assume response time of 0 to 4 seconds) in the legend (or at least add a unit to 0…4.

3) Please add some information on the temporal resolution of the fMRI to the methods. Volumes per "run" are specified, but it is unclear what the authors mean by "run" here.

4) L.78-79 "The nonhuman primates as an important animal model for auditory processing because of their social behaviour and cognition are similar to those of humans." If the authors insist on highlighting this, they should probably also highlight that NHP differs from humans in the likely most relevant dimension when it comes to auditory processing in that they are not capable of vocal learning.

5) Could authors explain what the rationale is for presenting in the main figures averaged fMRI data (Figure 2A) vs fMRI data for a single animal (Figure 3A). If Figure 2A is created for each animal separately, will we still see the same pattern of hierarchical processing?

6) Why did the authors choose a different number of clusters to determine local (cluster size>5) vs global novelty responses (cluster size>10)?

7) L. 427 "Interestingly, in the xY|xx condition, the global effect was also observed in MG, suggesting that the auditory regions in the subcortical regions may also process the large timescale auditory information." – unclear how the authors arrive at this conclusion.

8) "dB" should probably read "dB SPL".

9) We would encourage the authors to expand their discussion on previous literature a bit with reference to similar work – e.g. (Grimm and Escera, 2012, Cornella et al., 2012, Escera and Malmierca, 2014, Parras et al., 2017, Dürschmid et al., 2016).

---

## [Author Response]

Reviewer #1 (Recommendations for the authors):Line 37: Putative instead of putatively.

The word “putatively” has been replaced with “putative”. Please see Line 38.

Line 42: Neural circuits. I really read circuits to represent individual neuronal connections, or at the very least, small local neural pathways. As an fMRI and ecog study, that kind of resolution isn't available. I would recommend using neural networks.

All “neural circuits” used in the manuscript has been replaced with “neural networks”. Please see Lines 43, 99 and 167.

Lines 49-58: There is a great deal of discussion here about cortical levels. However this study includes data from the thalamus and IC, both of which are generally considered to be subcortical. I would suggest the authors define their terms more clearly, either by just using the term auditory pathway, or higher.

We agree that the introduction should also include subcortical regions in the hierarchy processing. Thus, we have revised it and added new references accordingly (Lines 50-58).

Line 78: Remove increasingly. Don't sell your model short, it is an important animal model and you tell us why right afterwards.

We have removed “increasingly” (Line 82). In addition, we have added the importance of the marmoset model in the *Discussion* (Line 470 to 487).

Line 78: I'm not sure exactly what the final layout will look like, but try to keep the figure close to the first reference to it. It's 4 pages after the fact right now, and I had to go looking.

We have moved Figure 1 to a better position, where it is suitable for paragraph arrangement.

Lines 77 and 78. Local-global is mentioned several times by this point, but is not defined nor is there a direct reference until 100 when it is finally detailed. Please put a reference on the first mention of it.

We have added both reference and figure right after the first mention of “local-global” (Lines 80-81).

Line 94: Again I prefer networks to circuits for this kind of macro level investigation.

We have changed the “circuits” to “neural networks” (Line 99).

Line 95. Use a shorter time scale, not a short time scale. 150ms isn't short compared to the response of a neuron. By keeping your terms relative, people like me can't complain.

We have used the “shorter time scale” instead of “short time scale” throughout the manuscript. Please see Lines 33, 101 and 455.

Lines 101-105, and especially the word alternatively at 103. The description of the stimuli is unclear. I cannot tell if x can be either 800 or 6000Hz, so that there are stimuli of 800 800 800 800 800 and 800 800 800 800 6000 as well as 6000 6000 6000 6000 6000 and 6000 6000 6000 6000 800 or only the former. Some rephrasing should easily tidy this up.

We thank the reviewer for the advice. We have rephrased the description of the stimuli. Please see Lines 109-110.

Line 153: PE1 and PE2 need to be defined in the caption text, not just in the figure image.

We have added the definition of PE1 and PE2 in the legend of Figure 1. Please see Lines 140-141.

Line 246: None of the MG or the cortical areas are "low-level auditory pathway". You could maybe get away with lower-level and keep it relative in the context of this study, but even then, you have IC which is lower than the MG, is not mentioned here, and is still not low-level as it's midbrain not brainstem. I'd recommend saying lower cortical and subcortical areas, or something to that effect. Be very aware that the IC does a lot of auditory processing, and probably is most closely analogous with V1 in the visual system, so this is the kind of error that can upset a lot of researchers.

We thank the reviewer for the suggestions. We have replaced “low-level” with “lower-level” throughout the manuscript. Please see Lines 266, 527.

Line 257: It took me quite a while to figure out what you meant by In the marmoset (J). Remember, we didn't do your study, so we don't know your labelling and won't because most people won't read your supplemental section. Say: In the marmoset individual, referred to as J, the increased…

We revised the description to “In the marmoset J”. Please see Line 301. We have added the definition of the two marmosets for the ECoG experiment (Lines 191-192).

Line 313: Remove neural. It implies you have a resolution that you don’t have.

We have removed the word “neural”. Please see Line 348.

Line 354: Monkey should be marmoset, be consistent in how you refer to your animals. Also, clearly indicate that J is the animal's identifier, and be consistent with that throughout the paper.

We have added the information of the animals’ identifier right after the first appearance in the ECoG results of these two marmosets. Please see Lines 191-192. In addition, we have replaced “monkey” with “marmoset” throughout the manuscript.

Line 363: Prediction not predictions.

We have replaced “predictions” with “prediction”. Please see Line 403.

Line 376: "suggesting that the prediction signal is updated on at the 2nd level sequence process". This is one of the most important results of the paper and is buried mid paragraph here. Show it off a bit more, be proud of your results, they're good.

We thank the reviewer for the positive comment. We have separated the paragraph, expanded the prediction results of the omission sequence, and discussed more its contribution to the predictive coding frame. Please see Lines 416-418, 463-468.

Lines 396 and 399: short and long temporal scales. I would really recommend defining your terms, calling them the shorter and longer time scales, and specify their lengths when you define them. Again, ecog and fmri just don't have what I would call short temporal resolution.

We have used “shorter and longer time scales” instead of “short and long time scales” throughout the manuscript. Moreover, the definition of the shorter and longer time scales has been added in the places where they were first mentioned. Please see Lines 101-103.

Line 412: Induce not induced.

The word “induced” has been replaced with “induce” (Line 491).

Line 417: Mouse not mice.

The word “mice” has been replaced with “mouse” (Line 502).

Line 420: Either "The neural response" or "neural responses to local intervals were".

The expression of “neural response to local intervals was” has been replaced with “neural responses to local intervals were” (Line 505).

Line 421: Declined is awkward here. Try attenuated.

The word “declined” has been replaced with “attenuated” (Line 506).

Line 427: Allows exploration of the…

We have replaced “allows to explore” with “allows exploration of” (Line 512).

Line 429: Subcortical areas instead of regions, to keep from repeating regions twice in the space of 5 words.

We have used “subcortical areas” instead of “subcortical regions” (Line 515).

Line 429: Bakay et al. 2018 and Robinson et al. 2016 both suggest this is likely as well, should probably reference them.

We thank the reviewer for reminding us. The articles are indeed relevant to the auditory processing in the subcortical areas and useful for the *Discussion* of our findings in the IC and MG. We have cited them accordingly in the revised manuscript. Please see Line 516.

Line 447: Remove neural. It's superfluous.

We have removed “neural”. Please see Line 540.

Line 465: Tones not tone.

We have used “tones” in the phrase “five identical tones”. Please see Line 560.

Line 469: May thus support that responses to prediction signals do not require.

We have used “responses of prediction signal do” instead of “response of prediction signal does”. Please see Line 564.

Line 473: … auditory cortex, which were elicited by…Also, elicited is a 5 dollar word and really awkward in this context, I'd just use found.

We have replaced “elicited” with “were found”. Please see Line 568.

Line 476: The early γ after following the omission onset.

We have used “the omission” instead of “omission”. Please see Line 572.

Line 482: Predictions. In conclusion our results from fMRI and ECoG recordings in marmosets reveal an account of hierarchical… Also, I'd recommend separating that in conclusion sentence out into its own paragraph.

We have modified the conclusion sentence and written it in a single paragraph. Please see Lines 578-581.

Reviewer #2 (Recommendations for the authors):1) Why are MGB and IC responses not shown in Figure S1B, and C? No activation would presumably also be interesting. Currently, it is unclear whether they were not activated, or whether it was not measured.

We didn’t show the MG and IC responses in Figure S1B and IC responses in Figure S1C, because there is no activation. We have supplemented the images of MG and IC in Figure 2—figure supplement 1C, Figure 3—figure supplement 1A and Figure 3A.

2) Figure 2A. Please explain t = 0…4 (we assume response time of 0 to 4 seconds) in the legend (or at least add a unit to 0…4.

The “t” in the fMRI parts of Figure 2-5, Figure 2—figure supplement 1A and C, Figure 3—figure supplement 1A, Figure 4—figure supplement 1A and B, Figure 5—figure supplement 1A, indicate the *t* values from the statistical parametric maps. After the estimation of within-subject one-way ANOVA design, we used a pair-wise *t*-test to produce all types of contrasts using the SPM toolbox, and then generated the statistical parametric maps, so that we can look for both “increases” and “decreases”. We added the statement “Color scale represents the *t* values” in the figure legends (see Lines 213, 274, 372 and 434). To illustrate the *t* value clearer, we also added more details about the statistical analysis in the method. Please see Lines 680-681.

3) Please add some information on the temporal resolution of the fMRI to the methods. Volumes per "run" are specified, but it is unclear what the authors mean by "run" here.

With our acquisition parameters, the temporal resolution of the fMRI data was 3.5s (one TR). The “run” means one EPI scanning run, which comprised 116 functional image volumes and lasted 3.5 s × 116 = 6 min 46 s. For the auditory task, one scanning run comprised a habituation block and three testing blocks. To make it clearer, we have added the temporal resolution of the fMRI data in the method, and revised the expression of “volumes per run” as “A scanning run comprised 116 functional image volumes”. Furthermore, in the fMRI and ECoG methods, we used “scanning run” and “recording run” instead of “run”, respectively. Please see Lines 602- 606.

4) L.78-79 "The nonhuman primates as an important animal model for auditory processing because of their social behaviour and cognition are similar to those of humans." If the authors insist on highlighting this, they should probably also highlight that NHP differs from humans in the likely most relevant dimension when it comes to auditory processing in that they are not capable of vocal learning.

We thank the reviewer for the suggestion. Marmosets are animals that can live in large extended family groups (Miller et al., 2016). One of the principal advantages of marmosets as a research model is their rich vocal repertoire and communicative vocal behaviors displayed both in the wild and in captivity (Eliades and Miller, 2017; Tsunada and Eliades, 2020). In fact, studies have suggested that marmosets have the capacity of volitional control of their vocalization and even some extent of vocal learning, even though the vocal learning of marmosets is not as flexible as humans and some birds (Takahashi et al., 2017). Therefore, we have revised the introduction to highlight the marmoset’s vocal learning ability (Lines 84-85).

5) Could authors explain what the rationale is for presenting in the main figures averaged fMRI data (Figure 2A) vs fMRI data for a single animal (Figure 3A). If Figure 2A is created for each animal separately, will we still see the same pattern of hierarchical processing?

We apologize for making the reviewer confused about Figure 2A and 3A. Both Figure 2A and 3A are group fMRI data, and the fMRI data for three single animals are shown in Figure 3—figure supplement 1A. Please see the statement in Lines 269-270: “Figure 3A; *p* < 0.01, FDR-corrected; Supplementary File 1c; individual marmosets: Figure 3—figure supplement 1A”. From Figure 2—figure supplement 1A and C, Figure 3—figure supplement 1A, we can observe that the patterns of hierarchical processing are consistent across the three marmosets, except the significant power of activation across the marmosets are a little different. To avoid confusion, we added “group-level fMRI” in the legends of Figures 2A, 3A, 4A and 5A.

6) Why did the authors choose a different number of clusters to determine local (cluster size>5) vs global novelty responses (cluster size>10)?

Compared to other brain areas in the auditory pathway, the structural sizes of IC and MG are relatively small. Therefore, in order to observe the significant fMRI activations caused by 1^st^-level novelty as much as possible, we used a threshold that cluster size of >5. To make the selection clear, we have added the reason for threshold selection in the method. Please see Lines 688-689.

7) L. 427 "Interestingly, in the xY|xx condition, the global effect was also observed in MG, suggesting that the auditory regions in the subcortical regions may also process the large timescale auditory information." – unclear how the authors arrive at this conclusion.

Indeed, there were no activations in MG in the xx|xY condition, where the 2^nd^-level novel responses were induced using xY as an internal template. However, the xY|xx novelty produced a violation at both 1^st^ and 2^nd^ levels, and we cannot eliminate the possibility that the global effect may occur in MG when the xx sequence was used as an internal template. In the present study, the fMRI cannot dissociate the local and global effects in xY|xx condition, and the ECoG doesn’t cover the subcortical areas. We thus proposed that future studies using single or multiple units recording should be performed in MG and IC regions in marmosets. We have rewritten the sentence to make the statement more accurate (Lines 513-518).

8) "dB" should probably read "dB SPL".

We have replaced “dB” with “dB SPL” (Lines 594 and 623).

9) We would encourage the authors to expand their discussion on previous literature a bit with reference to similar work – e.g. (Grimm and Escera, 2012, Cornella et al., 2012, Escera and Malmierca, 2014, Parras et al., 2017, Dürschmid et al., 2016).

We thank the reviewer for reminding us of the impressive work. They are indeed relative to the hierarchical auditory processing and helpful for enriching the discussion. We have cited them in the appropriate parts. Please see Lines 495-501.

References:

Cacciaglia, R., Costa-Faidella, J., Zarnowiec, K., Grimm, S., and Escera, C. (2019, Feb 1). Auditory predictions shape the neural responses to stimulus repetition and sensory change. Neuroimage, 186, 200-210. https://doi.org/10.1016/j.neuroimage.2018.11.007

Chao, Z. C., Takaura, K., Wang, L., Fujii, N., and Dehaene, S. (2018, Dec 5). Large-Scale Cortical Networks for Hierarchical Prediction and Prediction Error in the Primate Brain. Neuron, 100(5), 1252-1266.e1253. https://doi.org/10.1016/j.neuron.2018.10.004

El Karoui, I., King, J. R., Sitt, J., Meyniel, F., Van Gaal, S., Hasboun, D., Adam, C., Navarro, V., Baulac, M., Dehaene, S., Cohen, L., and Naccache, L. (2015, Nov). Event-Related Potential, Time-frequency, and Functional Connectivity Facets of Local and Global Auditory Novelty Processing: An Intracranial Study in Humans. Cereb Cortex, 25(11), 4203-4212. https://doi.org/10.1093/cercor/bhu143

Eliades, S. J., and Miller, C. T. (2017, Mar). Marmoset vocal communication: Behavior and neurobiology. Dev Neurobiol, 77(3), 286-299. https://doi.org/10.1002/dneu.22464

Miller, C. T., Freiwald, W. A., Leopold, D. A., Mitchell, J. F., Silva, A. C., and Wang, X. (2016, Apr 20). Marmosets: A Neuroscientific Model of Human Social Behavior. Neuron, 90(2), 219-233. https://doi.org/10.1016/j.neuron.2016.03.018

Takahashi, D. Y., Liao, D. A., and Ghazanfar, A. A. (2017, Jun 19). Vocal Learning via Social Reinforcement by Infant Marmoset Monkeys. Curr Biol, 27(12), 1844-1852.e1846. https://doi.org/10.1016/j.cub.2017.05.004

Tsunada, J., and Eliades, S. J. (2020, May 20). Dissociation of Unit Activity and Γ Oscillations during Vocalization in Primate Auditory Cortex. J Neurosci, 40(21), 4158-4171. https://doi.org/10.1523/jneurosci.2749-19.2020

Uhrig, L., Dehaene, S., and Jarraya, B. (2014, Jan 22). A hierarchy of responses to auditory regularities in the macaque brain. J Neurosci, 34(4), 1127-1132. https://doi.org/10.1523/jneurosci.3165-13.2014

Wacongne, C., Labyt, E., van Wassenhove, V., Bekinschtein, T., Naccache, L., and Dehaene, S. (2011, Dec 20). Evidence for a hierarchy of predictions and prediction errors in human cortex. Proc Natl Acad Sci U S A, 108(51), 20754-20759. https://doi.org/10.1073/pnas.1117807108